# Identifying actions to foster cross-disciplinary global health research: a mixed-methods qualitative case study of the IMPALA programme on lung health and tuberculosis in Africa

Yan Ding ,[1] Ewan M Tomeny ,[2] Imelda Bates ,[1] On behalf of The IMPALA Consortium

¹Centre for Capacity Research, Liverpool School of Tropical Medicine, Liverpool, UK
²Clinical Sciences, Liverpool School of Tropical Medicine, Liverpool, UK

**Correspondence to**
Dr Yan Ding;
yan.ding@lstmed.ac.uk

## ABSTRACT

**Objectives** To identify actions for fostering cross-disciplinary research (CDR) skills and collaborations in global health, and to produce recommendations for improving the design, implementation and management of cross-disciplinary global health research programmes.

**Design** Using a North–South global health research programme as a case study—and following an adapted framework—we conducted qualitative research using document reviews, semi-structured interviews (purposive sampling) and participatory observation. We used baseline survey findings to identify potential interviewees and tailor interview guides.

**Setting** Our case study was a 4.5-year (2017–2021) programme, namely, the International Multidisciplinary Programme to Address Lung Health and Tuberculosis in Africa (IMPALA). Led by a UK research institute, IMPALA spanned 22 partner organisations from 13 countries (10 in sub-Saharan Africa), and involved five research discipline groups: *clinical science*, *social science*, *health systems*, *health economics* and *policy/research capacity*.

**Participants** Thirty-one IMPALA members were interviewed (July 2018–November 2019), with interviewees evenly split by gender (16 female and 15 male) and by Global North/South institution (15 non-African and 16 African). Twenty-five (81%) were researchers, comprising 18 senior researchers (professors, readers, associate professors and senior lecturers) and seven early career researchers (assistant professors, lecturers, research fellows, postdocs, research assistants and PhD students). Twenty-four programme events were observed (September 2018–April 2020) and 49 documents were reviewed (December 2017–April 2020). All 66 IMPALA staff were sent the baseline survey, receiving 51 responses (43/56 researchers and 8/10 non-researchers).

**Results** Fourteen themes emerged, which suggested that CDR—while valued by many—is not universally understood, and the time it requires is often underestimated. We found that fostering CDR and managing tensions needs planning and continuous discussions and interactions. A shared vision with explicitly agreed goals and roles and active management of cross-disciplinary activities is essential.

## Strengths and limitations of this study

► We used an adapted published framework and a recent literature review to frame our data collection tools and analysis and have placed our findings in the context of current global knowledge concerning cross-disciplinary research (CDR).

► The credibility of our findings is strengthened from having used interview and observational data from diverse interviewees and events, corroborated by document analysis.

► Our study focused on a single cross-disciplinary global health research programme and its projects.

► We have enhanced the transferability of our findings by describing the complexity of the programme and the context within which the CDR took place.

► Our role as International Multidisciplinary Programme to Address Lung Health and Tuberculosis in Africa (IMPALA) members in conducting research on cross-disciplinary working in IMPALA may affect interviewees' responses, which we mitigated by ensuring confidentiality.

**Conclusions** Active planning, implementation and management of cross-disciplinary activities are essential for the success of cross-disciplinary global health research and should be separate from the primary research activities.

## INTRODUCTION

Bringing together researchers from multiple disciplines can lead to innovation and rapid production and dissemination of cross-disciplinary knowledge to solve complex global health problems.[1 2] Cross-disciplinary research (CDR) has been growing globally in popularity among researchers and funders because of its importance in addressing global health challenges.[1 2] 'CDR' covers three typologies: multidisciplinary, interdisciplinary and transdisciplinary research. In this article, we

will use the term CDR to mean research that combines concepts, methods and theories drawn from two or more disciplines.[3]

Existing evidence on fostering CDR is fragmented across disciplines,[4 5] making it difficult to find. There is increasing interest in understanding how to implement effective CDR and in the importance of team dynamics between researchers from disparate disciplines. CDR tends to be more complex than traditional types of research[6] and presents unique challenges,[1 3] such as problem definition, positioning in different disciplines[7] and coordination of effort.[8 9] Our previous literature review found that evidence about how to conduct effective CDR is primarily from high-income countries and may not apply to CDR in global health, where research is typically conducted through north–south collaborations.[3]

We used the International Multidisciplinary Programme to Address Lung Health and Tuberculosis in Africa (IMPALA 2017–2021)[10] as a case study to explore and reflect on practical actions for fostering CDR in north–south collaborations. IMPALA aimed to generate knowledge and implementable solutions concerning lung health and tuberculosis. Led by a Global North research institute, IMPALA had 22 international partner organisations from 13 countries and 10 in sub-Saharan Africa.

IMPALA explicitly used multidisciplinary approaches and spanned biology to policy.[11] It involved five research disciplines: *clinical science, social science, health systems, health economics* and *policy/research capacity*. Unusually, to promote fairness and overcome disciplinary hierarchies, the programme was framed around these discipline groups: each group initially received the same amount of funding and was represented on the management team alongside the three consortium directors. Each group had one PhD student and one Post-Doctoral Research Associate (PDRA) (figure 1), with equal training opportunities offered to all early career researchers (ECRs).

Our study has drawn on IMPALA as a whole and its two embedded projects (hereafter 'the two projects'): one combined *clinical science* and *health economics*; the other *health systems* and *social science*. This qualitative study explores the actions taken to foster CDR in the 'real-life' situation of a large programme (IMPALA). Our aim was to recommend actions that can be used to improve the effectiveness of future global health CDR programmes.

## METHODS

We adapted a previously published model of CDR collaborations, the 'Four-Phase Model of Transdisciplinary Research' (figure 2),[9] which describes objectives within each project phase (ie, *development, conceptualisation, implementation* and *translation*). We combined the development and conceptualisation phases into one 'planning' phase, since global health research activities in these phases are generally integrated.[12] The translation phase was not included because it requires long-term follow-up. Our literature review indicated that leadership and management strongly influence CDR effectiveness,[3] so these were added as a cross-cutting framework component.

### Data collection

Our primary source of data was semi-structured interviews, supplemented by a baseline survey, a document review and observations of events.

### Baseline survey

All IMPALA members were invited to complete a baseline survey (May–September 2018). This included individuals from the external scientific advisory panel, leadership and management teams, administrators and researchers/policy makers involved in the two projects. Participants were emailed an information sheet (online supplemental file 1) prior to beginning the online survey, and agreement to participate confirmed by the signing

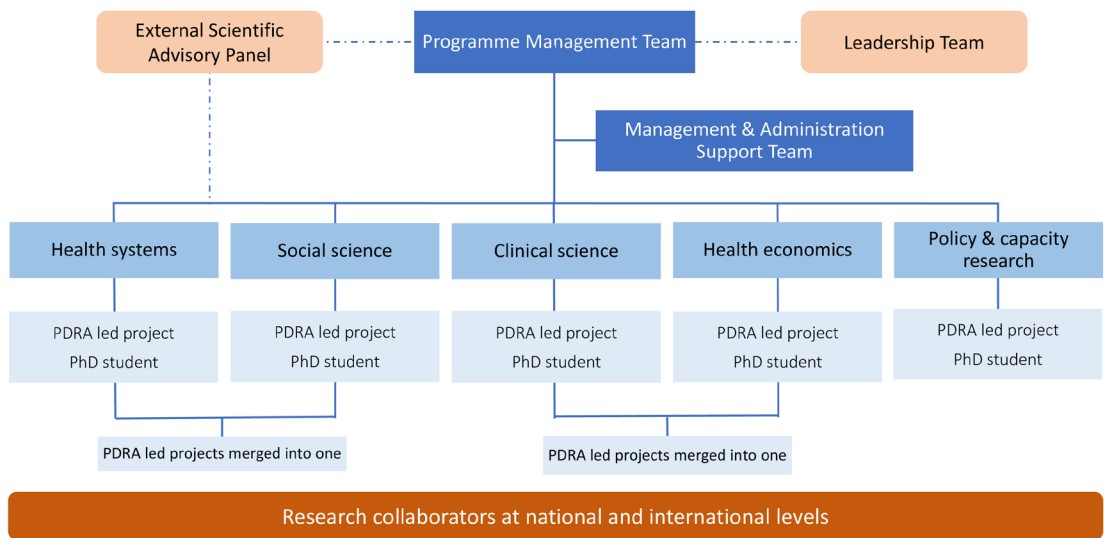

**Figure 1** The International Multidisciplinary Programme to Address Lung Health and Tuberculosis in Africa organogram.

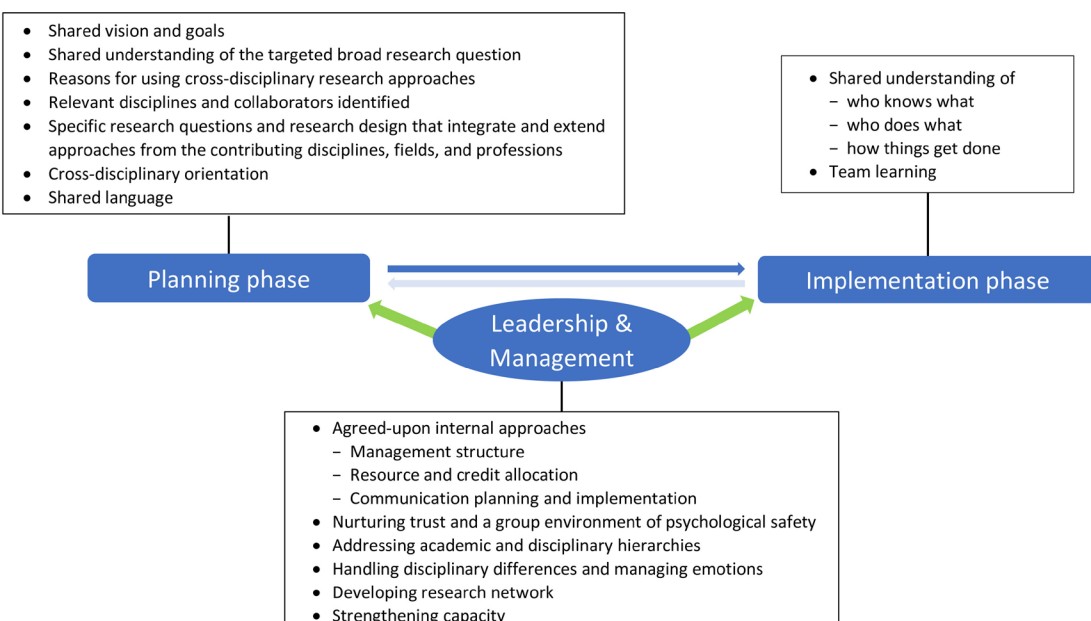

**Figure 2** The three-component framework for the cross-disciplinary collaborative research process used in this study (adapted from[9]).

of an online consent form (online supplemental file 2). The survey (online supplemental file 3) collected participants' personal information and their experience of, and confidence in, conducting CDR. The survey findings were used to identify potential interviewees and tailor interview guides.

## Semi-structured interviews

The interviews collected data on challenges and practical actions/solutions related to the fostering and conducting of CDR in IMPALA.

### Sample selection

Guided by our baseline survey data, 31 primary interviewees were selected using the IMPALA team directory as a sampling frame. Purposive sampling was used to maximise variation in roles, disciplinary backgrounds, career stages, gender, affiliated organisations and geographical locations.[13] These characteristics were primarily collected from baseline survey findings.

### Procedure of interviews

YD, an experienced social scientist with substantial experience of research interviews, carried out the interviews between July 2018 and November 2019. Interview questions (online supplemental file 4) were based on the adapted framework (figure 2) with probes informed by our literature review.[3] The interviewer asked neutral and open-ended questions without assumptions. Interviews were audio recorded and conducted in English, either in person or virtually. Participants were sent a project information sheet (online supplemental file 5) before their interview, with informed consent (online supplemental file 6) obtained in writing before each in-person interview

and via email for Skype interviews. For anonymity, each interviewee was assigned an identification number.

### Reflexivity

We used reflexivity throughout the interview process to improve the rigour of the data collection. We acknowledged that our role as IMPALA members in conducting research on cross-disciplinary working in IMPALA may have affected interviewees' responses. We attempted to mitigate this bias by reassuring participants of strict confidentiality and that our findings would be unidentifiable when reported. The interviewer transcribed the first four interviews to familiarise herself with the data and to reflect on the interview process for further improvement. The interviewer also had several debriefing meetings with IB—the senior researcher—reflecting on how to further improve interviews and on data analysis.

## Document review

Data were extracted from documents concerning the programme's *vision*, *goals*, *research questions*, *design*, *teams*, *interactions* and *outputs*, to understand the context of the programme and its projects, inform interview questions and cross-check findings from other data collection methods. Documents included the IMPALA website, concept notes, proposals, minutes/agendas from annual meetings and quarterly research updates.

## Observation of events

YD was a participant observer at IMPALA events involving cross-disciplinary issues including two annual meetings, monthly knowledge exchange meetings, training workshops and a 4-day field visit to Tanzania. After receiving oral consent from event participants, observation notes were entered in real time into a predesigned form

(online supplemental file 7) informed by the literature,[14] comprising sections on brainstorming the crossing of analytical levels, integration of disciplinary ideas, proposed/actual cross-disciplinary outcomes, information sharing, technical or emotional support, and challenges and setbacks.[14] Observation findings were used for refining interview questions and triangulating interview data.

## Data analysis

Interview data were coded, mapped and analysed using the framework (figure 2) with narrative summaries created through a combined inductive and deductive approach. This method used thematic synthesis through a 'constant comparison' method,[15] wherein themes and subthemes were identified throughout the coding, which were then adjusted iteratively by constantly comparing among them through reflection and analyses. In this way, the themes and subthemes were refined and integrated to form the basis of a coherent and explanatory descriptive narrative. Information from the document review and the observation forms that related to the narrative themes were summarised and compared with these narrative themes to triangulate the findings.

## Patient and public involvement

While the IMPALA programme involved both patients and the public, due to this study's specific focus on research practice, its design, conduct and reporting did not involve patients or the public.

# RESULTS

## Interviewee characteristics

Thirty-six interviews with 31 interviewees were conducted, each lasting 68–192 min. Five individuals were interviewed again after 1 year to identify changes in CDR in the two projects. Fifty-two percent of the interviewees (16/31) were female, and 16 were based at African organisations from 7/10 partner African countries. Twenty-five (81%) were researchers, comprising 18 senior researchers (ie, professors, readers, associate professors and senior lecturers) and 7 ECRs (ie, assistant professors, lecturers, research fellows, postdocs, research assistants and PhD students) (table 1).

## Survey, document reviews and observations

The baseline survey was sent to 66 IMPALA staff, with responses received from 43/56 researchers (77%) and 8/10 non-researchers (80%). Twenty-four events were observed over 20 months (September 2018– April 2020) and 49 documents were reviewed (box 1).

## Research results

Fourteen themes emerged from the findings, five for planning, three for implementation, and six for leadership and management (box 2). Interviewee's anonymised quotes are presented with their main role (researcher/non-researcher) and location (Africa/non-Africa).

### Actions that fostered CDR in the planning phase
#### Shared vision and goals

Interviewees identified that codevelopment of the IMPALA proposal between members from the Global

| | Items | Option | N |
|---|---|---|---|
| **Table 1** | **Interviewees' characteristics** | | |
| 1 | Role in International Multidisciplinary Programme to Address Lung Health and Tuberculosis in Africa (options not mutually exclusive) | A member of the external scientific advisory panel | 2 |
| | | A member of the leadership team | 4 |
| | | A member of the management team | 8 |
| | | A member of the management and administration support team | 3 |
| | | Other member working across IMPALA projects | 2 |
| | | A researcher or policy maker on the two projects (of those based in Africa) | 15 (8) |
| | | An IMPALA member who was based in Africa but not on the two projects | 5 |
| 2 | Gender | Female | 16 |
| | | Male | 15 |
| 3 | Location | African country | 16 |
| | | Non-African country | 15 |
| 4 | Primary disciplinary background | Medicine and clinical sciences | 18 |
| | | Humanities and social sciences | 10 |
| | | Others | 3 |
| 5 | Profession | Researcher/research leader | 25 |
| | | Non-research member | 6 |
| 6 | Academic rank (of the 25 researchers) | Senior researcher | 18 |
| | | Early career researcher | 7 |

**Box 1  Internal IMPALA documents used to provide background information for this study**

► The IMPALA website (https://www.lstmed.ac.uk/impala).
► IMPALA technical proposal.
► IMPALA team directory.
► The concept notes of all the eight research projects sitting under IMPALA.
► IMPALA publication guidelines.
► IMPALA data sharing, access and release policy.
► IMPALA data management guidelines.
► IMPALA communications plan.
► IMPALA kick-off meeting in 2017, annual meetings in 2018 and 2019, including:
  – Meeting schedule.
  – Attendees list and biographies.
  – Meeting slides.
► IMPALA technical reports, including
  – IMPALA 2017 report (covering the first 6 months of IMPALA).
  – IMPALA 2018 annual technical report.
  – IMPALA 2019 annual technical report.
► Research ethics application documents of the two case study projects, including research proposals and data collection tools.
► Quarterly updates by the four postdoctoral researchers working on the two case study projects (August 2018–December 2019, 20 documented updates in total).
► IMPALA year 1–3 joint outputs list.

IMPALA, International Multidisciplinary Programme to Address Lung Health and Tuberculosis in Africa.

**Box 2  Summary of the 14 themes which emerged from the findings**

**Five themes for planning phase**
► Shared vision and goals.
► Expectations of programme-level goals and success.
► Shared understanding of research questions and activities at the project level.
► Reasons for using cross-disciplinary research in International Multidisciplinary Programme to Address Lung Health and Tuberculosis in Africa.
► Cross-disciplinary orientation.

**Three themes for implementation phase**
► Shared understanding of roles and responsibilities.
► Reconciling individual expectations while navigating different contexts.
► Team learning.

**Six themes for management and leadership component**
► Communication planning and implementation.
► Nurturing trust and a group environment of psychological safety.
► Addressing disciplinary hierarchies through the management structure.
► Handling disciplinary differences and managing emotions.
► Developing research networks for possible future collaborations.
► Strengthening capacity.

South and North helped them reach a common vision. While this was time-consuming due to the large number of cross-disciplinary, interorganisational and geographically distanced members, several factors helped the process, including the existence of previous/ongoing collaborations and involvement in professional associations.

During the face-to-face start-up meeting, IMPALA members and the 22 participating institutions introduced themselves, and IMPALA's vision and strategic objectives were discussed. Specific goals for projects—and for IMPALA as a whole—had purposefully been left undefined by the management team so they could be codeveloped during this meeting. Interviewees reported finding this meeting useful for grasping the 'bigger picture' of IMPALA and for learning about one other.

### Expectations of programme-level goals and success

Interviewees had different expectations of IMPALA, depending on their seniority and disciplinary background. Senior clinical researchers tended to focus on the need to expand collaborations with partners. Two interviewees suggested that since many senior researchers had clinical science backgrounds, IMPALA provided more opportunities for clinical researchers to expand collaborations, compared with other programmes. Interviewees from non-clinical disciplines (eg, social sciences and health systems) were more focused on their existing projects and research quality. Senior researchers sought to enhance ECR's research skills, and ECRs were focused on generating outputs and building working relationships. Non-researchers focused on programme delivery and capacity strengthening in areas such as financial management, leadership and policy engagement. All interviewees reported expecting IMPALA to lead to new research questions and new funding. Observation data confirmed all of these findings.

Interviewees recognised the complexity of aligning project and programme goals. Two interviewees acknowledged the difficulty of collective prioritisation and proposed mapping the connections between programme and project objectives, possibly annually. One participant stated that, although seeking clarity around programme goals can facilitate members' engagement, balancing partnership development against addressing a large-scale broad research question with multiple disciplines is difficult.

### Shared understanding of research questions and activities at the project level

IMPALA's proposal outlined broad topics for research projects —with named leads and partners for each— while leaving specific research questions and activities to be developed during the start-up meeting. Project leads recognised that this allowed research questions to be based on the interests and experience of partners, and some expressed appreciation that programme leaders had not imposed personal priorities.

Researchers from both Global North and South were comfortable with this process, with Global South partners feeling they had driven the research agenda:

> I was looking at ways how I can also contribute rather than just passively engage in national meetings… we were there to conceptualise…what we want to do, … we got the research budget. (ID-21, researcher, Africa)

Others noted a risk of mismatch between programme and project goals and had difficulty narrowing research questions down from programme to project level:

> When you have… multiple perspectives that lead to such a broad potential for research questions that narrowing down and getting in some consensus can be quite difficult. (ID-1, researcher, Africa)

Two project teams addressed this differently:

One developed research questions based on a baseline assessment conducted during joint field trips with local research and implementation teams, enabling them to develop locally important, high-priority research questions. To address these questions, they drew on methods from their two core disciplines, indicating some complementarity in their disciplinary paradigms such as theories (eg, pragmatic health systems thinking, community engagement and empowerment), research methods (eg, quantitative research methods for health systems data, qualitative research methods to understand the quantitative data further and participatory action research approaches) and standards (eg, pragmatic and efficiency, local ownership, feasibility and acceptability, and sustainability). The benefits of having one project integrating two disciplinary components appeared clear to this team from the outset.

The other project team initially generated their research questions independently within each of their two disciplines and then merged the projects through discussions and negotiation, which were 'initially uncomfortable' (ID-13, researcher, non-Africa). One researcher believed 'practical efficiency in terms of time and data collection' (ID-9, researcher, Africa) of this approach to have been the main advantage of merging the two disciplinary research projects into one.

### Reasons for using CDR in IMPALA
IMPALA took a CDR approach as it was felt its broad research question—that is, *to address lung health and tuberculosis in Africa*—required inputs from multiple disciplines, and programme leaders recognised that everyone had a role in ensuring research findings informed policy. Interviewees considered CDR as one of the 'most effective ways to generate the best possible outputs and outcomes' (ID-13, researcher, non-Africa) since it 'enables appropriate generalisation of research outcomes' (ID-15, researcher, non-Africa). Several interviewees mentioned that multidisciplinary research was a funder's requirement;

however, one cautioned 'don't just do [CDR] for the sake of it' (ID-14, researcher, non-Africa).

While most senior researchers recognised the importance of CDR, most interviewees (researchers and non-researchers) had not participated in explicit discussions on what actions would be needed to conduct CDR.

> A lot of the challenges is people are so busy doing their own things that they forget that that is what needs to happen. (ID-12, researcher, non-Africa)

The IMPALA programme included a postdoctoral researcher (YD) dedicated to investigating cross-disciplinary working. The definitions of multidisciplinary and interdisciplinary research and CDR were presented to IMPALA members during the second IMPALA annual meeting, prompting discussions and clarifications. However, interview findings suggest such clarifications would have been useful earlier, alongside discussions on prespecified goals/methodologies concerning cross-disciplinary working.

### Cross-disciplinary orientation
Observations clearly indicated that IMPALA members valued understanding more about each other and their disciplines especially within a group environment of psychological safety while highlighting the value of clarifying disciplinary boundaries to prevent conflicts.

Having inputs from colleagues with various disciplinary backgrounds at the planning phase and arranging formal time for candid conversations on research questions and design were viewed by interviewees as critical. A programme leader and a researcher highlighted potential tensions in cross-disciplinary working and the need for maintaining 'discipline uniqueness'. The process of defining and clarifying research goals among disciplines was considered to have helped clarify disciplinary boundaries:

> After the goals are fixed and then each goal somehow belongs to certain disciplines…relate data to that goal and then deal with the data, publication, all those things followed. (ID-15, researcher, non-Africa)

### Actions that fostered CDR in the implementation phase
#### Shared understanding of roles and responsibilities
Collaborative working was facilitated by a shared understanding of the roles and contributions of different disciplines and partners, along with an appreciation that successful cross-disciplinary collaborations require complementarity rather than competition. This helped team members to overcome 'fighting for space' and 'struggling for context leadership' (ID-22, researcher, Africa). Several interviewees noted the importance of research administrators in helping to understand responsibilities:

> Because we [administrators] are that sort of hub in the middle, and we do oversee everything. We can sort of speak on behalf of the project and say that this

isn't working and have a bit of input in that way. (ID-3, non-researcher, non-Africa)

Several interviewees had not had open discussions about roles and responsibilities, with one suggesting that roles were defined by one's job description and another explaining that 'as a member of the team you naturally know your strengths and therefore role' (ID-5, researcher, Africa). Another interviewee highlighted that assumptions regarding roles and responsibilities had the potential to cause confusion and needed open discussions:

I increasingly think the best way to have good, harmonious, collaborative relationships is to be really upfront about roles and responsibilities. To do that first so that there is no confusion after. (ID-9, researcher, Africa)

One interviewee suggested that jointly developing a work plan containing explanations of responsibilities alongside a clear timeline could help to clarify roles.

### Reconciling individual expectations while navigating different contexts

Several interviewees advocated for open discussions on roles, suggesting such discussions were important because people were at different career stages with different experiences, cultures and academic systems, which could cause mismatched expectations of one another's roles. This led to, for example, disagreements on the time spent in the research sites and responsibilities for research coordination. Clarifying roles and having a host country/institution coordinator was thought to be essential in avoiding these issues.

Regular cross-disciplinary project update meetings, along with individual conversations to provide performance feedback to ECRs (including those with different disciplinary backgrounds), were said to be useful by both ECRs and senior researchers. Role modelling was also identified as important in encouraging ECRs to continuously explore other disciplines:

Seniors and line managers say, 'You should go to this. Think about this…' So, it does need people, at a senior level, to think broadly and encourage that. (ID-23, researcher, non-Africa)

Support across disciplines was valued during project implementation, for example, when developing questionnaires and collecting and analysing data, and several senior researchers called for more thought on how to provide supportive supervision:

Perhaps we didn't think hard enough about how to support the projects and who should be supporting the projects and in what way. (ID-9, researcher, Africa)

### Team learning

The importance of individuals' ability to blend disciplinary edges was raised by an interviewee, and many others shared their approaches to understanding other disciplines. Senior researchers also encouraged colleagues to consider broadening the scope of their work and skillset through formal cross-disciplinary training, mutual learning and joint supervision in other subject areas. One month after the interviews, monthly knowledge exchange meetings were initiated to improve cross-disciplinary learning and communication, according to our observation and review of programme documents.

### Leadership and management
#### Communication planning and implementation
New IMPALA members appreciated their one-to-one induction meetings with key researchers and administrators. Joint site visits by members from the Global North and South were helpful in forming relationships and in promoting cross-fertilisation. Face-to-face meetings were valued for facilitating the design, prioritisation and development of both research projects and teams, especially concerning developing methods and budgeting. Interviewees said that virtual meetings and email communications worked well and were useful, though several raised issues with internet connections. Effective planning to maximise the availability of team members was highlighted:

What I usually do is to inform them early enough because they have lots of responsibilities…After they have considered then you block the time… With multi-disciplinary, it needs proper planning, especially on timing. (ID-26, researcher, Africa)

Many senior researchers often had long working relationships with country partners. To help ECRs to build mutual understanding and to develop research networks, regional meetings for ECRs across disciplines were suggested.

Several interviewees suggested that having access to other teams' materials and outputs could have improved cross-disciplinary understanding. A common platform for document and information sharing was subsequently established. Interviewees further proposed that cross-disciplinary communications should be expanded. Accordingly, the monthly knowledge exchange meetings were expanded beyond ECRs to include administrative staff, in-country partners and researchers beyond IMPALA's core team.

Interviewees wanted more time to develop mutual understanding in CDR and to create a sense of ownership. One interviewee reflected 'we need to have some more recognition of the need for time for some of the processes and the collaborations to work for the future' (ID-11, researcher, non-Africa). Another recommended taking time to learn about each other's experiences and expectations, ways to successfully collaborate and for joint preparation of project tools (eg, databases).

According to several interviewees 'there are inevitable delays in starting' (ID-9, researcher, Africa), for example, in funding release (6 months), international

staff recruitment (5–8 months) and ethics approval (7–8 months). Interviewees described how they felt the need to focus on outputs, although 'would have loved to have used those six months to think about how we prepare these disciplines to work together' (ID-11, researcher, non-Africa). One interviewee highlighted the importance of prioritising internal communication even within tight timescales, arguing 'sometimes prioritising a two hour meeting to make sure everyone's on the same page and understanding things in the same way is equally important as papers and research outputs' (ID-11, researcher, non-Africa).

### Nurturing trust and a group environment of psychological safety
Two senior researchers, three ECRs and three non-researchers noted that IMPALA management had helpfully promoted involvement and empowerment of ECRs and non-researchers, and two ECRs appreciated the space and freedom their line managers had given them to lead projects.

There were three other suggestions offered by interviewees for nurturing trust: (1) treating everyone equally through 'flat management': 'I very strongly believe in flat management, a structure everybody is equal. If I have a research meeting in my team, they all know we are equal. If they have something to say, they are all happy to say, and confident to say it' (ID-2, researcher, non-Africa); (2) building trust by delivering on commitments (mentioned by two researchers and one non-researcher): 'To build trust you need to deliver… I think that's important, showing that you want to do your best. Then by reflection they don't want to let me down, so they deliver, and that's how you build trust, I think' (ID-4, non-researcher, non-Africa); (3) being transparent and learning from mistakes: 'Transparent, I think building trust… Also within trust and team, you have to allow mistakes… Accepting and also sitting together and see how we can handle it next time' (ID-21, researcher, Africa).

### Addressing disciplinary hierarchies through the management structure
According to three interviewees, disciplinary hierarchies emerged when one discipline's work depended on another's. For example, when one discipline's research questions and analysis relied on another's data generation, the latter may perceive their research activities should be prioritised over the former. Despite both projects having been allocated equivalent resources at the start of IMPALA, perceived imbalances arose. Five interviewees suggested that since clinical aspects were the primary interest of several IMPALA leaders, this may have inadvertently contributed to disciplinary hierarchies. Furthermore, several interviewees found the equal allocation of resources limiting, potentially hindering the effective answering of some research questions. Two interviewees further noted that since studies were highly interconnected at the operational level, strict drawing of financial boundaries between projects could at times 'lead to tensions' (ID-1, researcher, Africa).

Following the initial equal allocation of resources, a degree of renegotiation continued throughout IMPALA's lifetime though some members questioned the success of this process. One remarked that 'an alternative approach may be to develop the budget based on justified activities' (ID-15, researcher, non-Africa).

### Handling disciplinary differences and managing emotions
At times, the different approaches and priorities of disciplines led to some disagreements. Overall, the group which combined clinical science and health economics was perceived as predominantly output-driven, whereas the humanities and social science group appeared primarily focused on processes, consultation and discussions. We observed frustration within cross-team project meetings and programme management meetings particularly in the first year of the programme; this observed frustration was confirmed in several interviews. One interviewee from the management team reflected: 'we probably hadn't paid enough attention to the need for the process [of discussions between the management team members] because it 'requires sustained effort to balance the natural priority of an individual's discipline against that of multiple disciplines' (ID-11, researcher, non-Africa). Two interviewees suggested that time spent discussing managerial and logistics issues could have been more productively spent on research activities and constructive management of disciplinary disagreements.

Several interviewees described encountering emotional challenges most frequently caused by disciplinary differences and some identified having needed dedicated meetings to manage emotions in a professional environment. One interviewee commented that their previous working relationships and sense of responsibility had helped to make these conversations possible.

Such conversations resulted in real-time adaptations to the programme to enhance cross-disciplinary relationships. For example, monthly directorate and management team meetings were merged, and a rotational system for the management meeting chair was instigated whereby each discipline lead and the consortium directors took turns in chairing. Handovers between meetings were supported by the programme management and administration support staff. Actions to promote more effective cross-disciplinary collaborations were also identified through a 1-hour consortium-level group exercise during the second annual meeting. This meeting was led by our research group on fostering CDR, and included small-group discussions with consortium members from a mix of disciplines, seniority, organisations and research teams. These actions were documented through a report with feedback from consortium members. Reviewing uptake of these actions became a standing item at management meetings. These changes were viewed as positive by several interviewees.

*Developing research networks for possible future collaborations*

Interviewees emphasised the programme's many good working relationships between different partners across Global North and South and noted the considerable benefit from strong previous relationships of key leaders. The importance of enabling the development of such research networks was a repeated theme from interviews:

> I think it [IMPALA] has really done a great job bringing great collaborators in terms of Africa with Liverpool, countries that are involved. I think it's really an interesting network and it has brought us together, many collaborators. People have never even met. (ID-21, researcher, Africa)

Two interviewees reflected that project activities had helped build up trust and develop research networks:

> I hope my work […] will let them [current IMPALA members] say that 'he would actually put the neck on the line and physically help you. Get him on board.' (ID-2, researcher, non-Africa)

*Strengthening capacity*

Several approaches to capacity strengthening were identified through interviews and corroborated by internal documents. These included:

► Training workshops for those with different disciplinary backgrounds from the training subjects (eg, training on social science research methodologies, policy engagement, statistics and spirometry).
► Coaching through team meetings and one-to-one discussions (eg, two interviewees emphasised that discussions with a statistician catalysed research).
► Mentoring ECRs and providing them with platforms at high-level international meetings (eg, the UN General Assembly).
► Learning through peer support and reflection was mentioned by ECRs, senior researchers and non-researchers:

> I feel like I'm definitely learning a lot… It's nice working so closely with […] and she's able to delegate things to me as and when they come up. (ID-3, non-researcher, non-Africa)

Capacity strengthening also involved administration and field teams:

> My ideal world would be a world where everyone can do it because that's capacity building in-country. And it is not just the research, it's the admins. (ID-4, non-researcher, non-Africa)

## DISCUSSION

We adapted and expanded a published framework to underpin our research. Our findings emphasise that CDR programmes require careful planning, implementing and managing, and we have identified actions to promote CDR including some that have not previously been published.

## Actions in programme planning to foster CDR

### Clarity in defining 'CDR'

Similar to other studies, we found a lack of agreement on defining multidisciplinary, interdisciplinary and transdisciplinary research.[16] Our findings demonstrate that explicit discussions concerning both definitions and what CDR means in practice are critical in the planning phase.

### Managing expectations and harmonising goals

Participants had different expectations about being involved in CDR and highlighted the importance of negotiating a clear shared vision, taking into consideration individuals' expectations.[17] To harmonise goals, frequent discussions and interactions such as information sharing can be helpful[3 18] and need to be more frequent and intensive than in monodiscipline research.[9] Our findings shed light on tensions that can arise early in CDR, including balancing flexibility and acceptance that not all aspects of the research could be initially 'nailed down', with developing a common understanding of the goals.

As with previous studies, IMPALA's participants recognised the importance of a common conceptual framework for outlining the vision, objectives and organisational structure for showing the contributions of each discipline[19] and to guide collaborations.[17 20] Furthermore, evidence suggests that having explicit knowledge integration goals for CDR is helpful.[20 21] IMPALA's conceptual framework was strengthened during the programme, for example, by taking account of local contexts (achieved through joint field trips and discussions), by codeveloping research questions and by drawing methods from relevant disciplines.

## Actions in programme implementation to foster CDR

Our findings reflect previous studies which suggest that cross-disciplinary relationships flourish if they are prospectively planned and actively monitored.[3 9] This is best managed separately from activities that focus on research outputs since fostering cross-disciplinary relationships requires its own planning and activities,[22] specific monitoring indicators and mechanisms for collecting data against the indicators.[23]

## Management actions to foster CDR

### Development of research collaborations and networks

Our study revealed important findings concerning management strategies for encouraging equitable partnerships, fostering CDR and reconciling individual expectations. These included involving northern and southern partners in codeveloping a shared vision and goals, designing project-level research questions and activities, and strengthening capacity in line with a baseline capacity assessment.

### Allowing time to promote cross-disciplinary activities

Our research also identified that researchers lacked sufficient time to successfully engage in discussions and processes to promote cross-disciplinary activities. Building in adequate time and funds for this throughout programmes is critical and may necessitate a shift in research planning as well as an understanding among research funders that such allocations are essential. Areas which could have benefited the most from additional time investment included the development of shared vision and goals, having inputs from colleagues with various disciplinary backgrounds at the planning phase, arranging formal time for candid conversations on research questions and design, development of mutual understanding and a better understanding of the processes of collaboration.

Lack of time for active consideration and management of activities to promote cross-disciplinary working is closely linked to lack of effective communication among programme members to bridge across disciplines.[24 25] While less of a consideration in monodisciplinary research, cross-disciplinary researchers must build mutual understanding and discuss acceptable ways forward.[26–29] Differences across disciplines can be vast and include philosophical,[25 30 31] measurement standards,[26] framing of concepts,[32] attitudes to theory and practice,[26] the use and understanding of terminology,[24 25 30] and expectations of communication and etiquette.[24 26] Interviewees proposed that cross-disciplinary communications should include all team members. This requires an agreed internal communication plan, administrative support and an electronic communication platform. Other studies have also highlighted the importance of an accessible space to document programme work and decision making.[33]

### Programme adaptations to address hierarchies and tensions

Our framework specifically recognised 'nurturing trust and a group environment of psychological safety', 'communication planning and implementation' and 'team learning' in CDR as important because of possible emotional issues associated with ownership, territoriality, academic and discipline hierarchy, and disciplinary differences. Similar to previous studies, our findings identified CDR-related emotional issues (particularly around power and hierarchy) and disagreements in disciplinary approaches.[17 34 35] IMPALA took measures to mitigate such frictions, including providing equal funding and training opportunities, and adjusting the programme's

**Table 2** Recommendations for the planning, implementation and management of cross-disciplinary global health research

| Research phase | Recommendations |
| --- | --- |
| Planning phase | ▶ Allocate adequate time to develop a shared vision and goals, including:<br> – Codesigning of programme goals.<br> – Aligning individuals' expectations and projects' aims with the programme-level goals.<br> – Involving all partners in proposal development, maintaining flexibility, considering individual interests and disciplines.<br> – Justifying and communicating the cross-disciplinary approaches to be adopted and reflecting cross-disciplinary processes in an action plan.<br> – Developing and communicating a shared understanding of the roles, responsibilities and potential contribution of disciplines and partners.<br>▶ Negotiate disciplinary boundaries when necessary.<br>▶ Assess and strengthen in-country teams' capacity in CDR and maintain clear plans for the involvement of in-country teams in decision-making processes. |
| Implementation phase | ▶ Jointly develop and pre-agree on internal approaches of working across disciplines, including communication, data access and management, publication policy and credit allocation.<br>▶ Track the implementation of cross-disciplinary processes with preagreed indicators and review and respond accordingly. |
| Leadership and management | ▶ Rotate chairs for programme management meetings to ensure prominence of all relevant disciplines and with a process for handover and preparation between meetings.<br>▶ Define and agree on transparent programme-level mechanisms for strategic decision making.<br>▶ Develop a programme-level leadership and management plan to deliver the cross-disciplinary outputs and outcomes, including regular review of tracking indicators.<br>▶ Agree on roles and responsibilities and accountabilities, and communicate these clearly to all programme members, making it explicit that every role is important in cross-disciplinary research (ie, not just researchers).<br>▶ Support an open culture of raising concerns and putting mechanisms in place for requesting support and responding to requests.<br>▶ Establish mechanisms for early identification of tensions and for reflecting on and flexibly resolving differences and conflicts.<br>▶ Provide opportunities for joint learning and knowledge exchange across disciplinary boundaries, especially methods and approaches (eg, monthly knowledge exchange meetings).<br>▶ Identify a platform for joint sharing and updating of documents. |

management structure. These included merging monthly directorate and management team meetings, having a rotating chair for management team meetings, initiating monthly knowledge exchange meetings for mutual learning and cross-disciplinary communications, and creating a common platform for document and information sharing. In addition, emerging findings from our study on CDR were presented at management team meetings and summarised in quarterly bulletins for all IMPALA members so that they could inform subsequent programmes.

## Strengths and limitations of the study

While our research team were not involved in decision making at the programme and project levels, as members of IMPALA, we had ongoing access to programme colleagues and documents, along with frequent opportunities for informal discussions. Nevertheless, we were conscious throughout that the conducting of our real-time investigation into the process of CDR within IMPALA may have influenced interviewees' responses. We therefore ensured that our interviewees understood that their participation was voluntary, that data would be handled confidentially and that our findings would be reported anonymously. To enhance trustworthiness, we used maximum variation sampling to enhance representation of the study population and saturation was achieved. The credibility of our findings is strengthened by having used multiple research methods, and by gaining multiple perspectives, which included research and administrative staff and non-academic partners across organisations and countries.[36] Although our study focused on a single cross-disciplinary global health research programme and its embedded projects, we have enhanced the generalisability of our findings by describing the complexity of the programme and the context within which the CDR took place.

## Three-component framework on CDR

Using our 'real-life' findings we adapted and expanded a published model of cross-disciplinary collaborative research processes[9] to create a framework useful for collecting and analysing multi-source and multi-perspective data on CDR in real-time. A new component of the framework emphasised the importance of leadership and management in CDR processes. We would recommend further adaptations to the framework to include a rationale for the components and to expand the 'shared understanding of who knows what, who does what, and how things get done'.[9] In addition to being useful for future research on CDR, our framework could be used to guide the design of cross-disciplinary programmes since it has practical applications across the three programme components of planning, implementation and management (figure 2).

## Recommendations

Based on the findings from our study, our adapted framework and our knowledge of the current literature, we have developed recommendations for planning and implementing future CDR in global health to improve the effectiveness of CDR processes from the outset (table 2).

**Acknowledgements** We are grateful to all participants who agreed to take part in this case study and provided their valuable information and insight. We are also thankful to the IMPALA central administration team and the Centre for Capacity Research administration team, Annmarie Hand, Elly Wallis, Lorelei Silvester and Zena Parker, who made the necessary logistical or administrative arrangements to allow a smooth data collection process. We also thank our Research Impact and Knowledge Translation Officer, Susie Crossman, for her support in reviewing and editing the manuscript. We are grateful to Martina Savio, Dr Angela Obasi and Professor Stephen Bertel Squire from the IMPALA Management Team for their review of an earlier draft.

**Collaborators** The IMPALA Consortium collaborators : Emmanuel Addo-Yobo, Brian Allwood, Hastings Banda, Amsalu Binegdie, Asma El Sony, Adegoke Falade, Jahangir Khan, Maia Lesosky, Bertrand Mbatchou, Hellen Meme, Kevin Mortimer, Beatrice Mutayoba, Louis Niessen, Nyanda Elias Ntinginya, Jamie Rylance, Miriam Taegtmeyer, Rachel Tolhurst, William Worodria, Heather Zar, Eliya Zulu, Lindsay Zurba and S Bertel Squire

**Contributors** YD: conceptualisation, methodology, data curation, writing (original draft preparation); EMT: methodology, writing (reviewing and editing). IB: conceptualisation, methodology, writing (reviewing and editing), supervision and acting as guarantor. All authors read and approved the final manuscript.

**Funding** This research was funded by the National Institute for Health Research (NIHR) (project reference 16/136/35) using UK aid from the UK government to support global health research. The views expressed in this publication are those of the authors and not necessarily those of the NIHR or the UK Department of Health and Social Care.

**Competing interests** None declared.

**Patient and public involvement** Patients and/or the public were not involved in the design, conduct, reporting or dissemination plans of this research.

**Patient consent for publication** Not applicable.

**Ethics approval** This study involves human participants and was approved by the Liverpool School of Tropical Medicine's Research Ethics Committee (reference: 18-031). Participants gave informed consent to participate in the study before taking part.

**Provenance and peer review** Not commissioned; externally peer reviewed.

**Data availability statement** Data are available upon reasonable request. The present study includes selected quotes to represent the content and themes across interviews. While requests for additional de-identified transcripts can be made to the authors, full interview transcripts cannot be made available as to do so would compromise anonymity and violate the terms of our ethical approval.

**ORCID iDs**
Yan Ding http://orcid.org/0000-0002-8439-9682
Ewan M Tomeny http://orcid.org/0000-0003-4547-2389
Imelda Bates http://orcid.org/0000-0002-0862-8199

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
