## [Reviewer comments · BMJ Open]

ARTICLE DETAILS

TITLE (PROVISIONAL)	Identifying actions to foster cross-disciplinary global health research: a mixed-methods qualitative case study of the IMPALA programme on lung health and tuberculosis in Africa
AUTHORS	Ding, Yan; Tomeny, Ewan; Bates, Imelda; on behalf of, The IMPALA Consortium

VERSION 1 – REVIEW

REVIEWER	Reeves, Julie University of Southampton, Centre for Higher Education Practice
REVIEW RETURNED	07-Nov-2021

GENERAL COMMENTS	The paper makes a unique contribution to our understanding of cross-disciplinary research through its qualitative research into a global health focused research group and by drawing out Southern and Northern perspectives. It will be useful for anyone embarking on a cross-disciplinary and international research projects, and has potential reach beyond health related issues. The framework provides researchers with a valuable approach for structuring the design and conduct of cross-disciplinary work – the recommendations (Table 2) are especially helpful. The findings and discussion all provide important insights into how the complex and specific challenges of cross-disciplinary work can be identified and mitigated. It is great to have qualitative research on cross-disciplinary research activity and that explores both the joys and challenges this kind of research brings. The authors have done the sector a service through their exploration of matters that affect the research but are seldom disclosed, yet cause delay and disruption. The way the project team modified its approach in the light of the problems and feedback, will provide reassurance to similar project teams not only in the health sector, I believe. There are several areas where the paper would benefit from further detail and examples to illustrate certain points that have largely been described and where the qualitative comments do not convey sufficient information or evidence. There are a number of occasions where an 'i.e.' or 'for instance/example' would make the case convincing for the reader (please see below). Also, in view of the number of sub-headings the paper could better sign-post/remind the reader of the key points at the beginning of main sub-sections. To strengthen the paper, the reviewer suggests the following (see BMJ Open page numbers):
--

- The paper explicitly draws on an 'adapted published framework', which is a strength. However, does this framework have a name? If so it would be helpful to name this framework in the body text for the reader. Further in the text (page 22 line 22) it may be useful to either name the framework or to repeat the stages of 'development, concept, implementation and translation'.
- Page 11, line 7: Text reads: 'Ten themes (7 themes concerning the Planning and Implementation Phases; 3 concerning Leadership and Management) emerged from the findings.' The themes that follow and the information in brackets is confusing. Is it possible to link the themes to the sub-headings more precisely? For instance, 5 for Planning, 3 for Implementation, 6 for leadership and management, as this will correspond directly to the sub-headings that follow. Although there seem to be 14 sub-headings (or themes) and it is not clear how they relate to the 10 themes. There is a need for greater clarity here, which should only require minor changes to the text. In addition, it may be that these sub-headings would benefit from an opening statement to orientate the reader.
- Page 13, line 21 – 23: Text reads '....indicating some complementary in their disciplinary paradigms such as theories, research methods, and standards.' Is it possible to be more explicit and offer an example of which theories, methods and standards? As the point of 'complementarity rather than competition' is important in this paper, an example would illustrate the authors' point more robustly.
- Page 19, Line 58: Sub-section: 'Handling disciplinary differences and managing emotions.' The discussion of emotions in this section is not very strong, although emotional issues are summed up nicely in the Discussion. Can the authors' supply further detail and insight in the section, i.e. how did these emotions manifest themselves (as serious argument or minor irritation)? What was their level of intensity?
- Page 20, line 48: Text reads '...cross-disciplinary collaborations were also identified through a group exercise...' Is it possible to indicate what kind of group exercise, such as what it involved etc? There are many aspects in the paper that are likely to be a source of interest, which the authors should not miss.
- Heading 'Developing research networks', page 20, line 55: Is this section about networks or networking relationships? Suggest the authors review for precision, as the quotations do not seem to explicitly relate to networks.
- Page 22, line 54. Is there any evidence you could cite within the text (especially in view of the literature review) that draws out the comparison with mono-disciplinary research?
- Page 23 line 9, Text opens with: 'As with previous studies....' are there any notable ones that could be mentioned in the text?
- Sub-section: 'Allowing time to promote cross-disciplinary activities', Page 24. Are the authors able to guide the readers as to how much time (even if only notional) might be required?

Possible Typos:

- Page 24, end of line 47 – There or These?
- Reference 17 – Nursing Outlook?

The paper is accessible, makes valuable use of qualitative data and contains good diagrams. I found it an enjoyable read – thank you

REVIEWER	Fàbregues, Sergi Universitat Oberta de Catalunya
REVIEW RETURNED	28-Nov-2021

GENERAL COMMENTS	While this is an interesting study with relevant findings, it has several major problems, particularly concerning the study methods, leading me to recommend rejection. These problems are summarized below.  - The study objectives are unclear. The statement on page 4, lines 49-49, that “our study explores the ‘real life’ (...)” is insufficient and not connected to the study methods. What are the objectives of the survey? What are the goals of the semi-structured interviews? Therefore, a more precise description of the study objectives and a more explicit linkage between these objectives and the methods are needed. - The description of the study methods lacks detail and clarity. For example, how was the survey data analyzed? How were the findings data used to identify the potential participants and build the interview guides? What was the rationale for using maximum variation sampling? Who carried out the interviews, and when were they conducted? What information did the information sheet contain, and why was it provided to the participants? In what ways could the information sheet have affected the interview findings? How was reflexivity during the interview managed? What criteria were followed to ensure trustworthiness? How were the data from the different sources triangulated? What was the rationale behind triangulation: validating the findings from the various sources; seeking complementarity? In what ways was the framework used to analyze the data? - The authors used a mixed methods design to carry out the study. While the authors don’t need to use the mixed methods jargon to describe the methods, they could use the guidelines suggested in that literature to improve the study reporting. Such guidelines include reporting a rationale for using several data sources, describing the type of design used, and making explicit how the integration between the data sources was carried out. - The authors cite a source on narrative analysis (Riessman, 2008) to justify the type of qualitative analysis, although the study was not based on a narrative approach. What did “thematic synthesis” imply in the study?
--

VERSION 1 – AUTHOR RESPONSE

Reviewer 1 – Julie Reeves – University of Southampton

6	R 1	Methods	The paper explicitly draws on an ‘adapted published framework’, which is a strength. However, does this framework have a name? If so it would be helpful to name this framework in the body text for the reader. Further in the text (page 22 line 22) it may be useful to either name the framework or to repeat the stages of ‘development, concept, implementation and translation’.	We have added the name of the existing framework which our framework builds upon, which is the ‘Four-Phase Model of Transdisciplinary Research’. This framework is also cited when first mentioned in our Methods, paragraph 1.
7	R 1	Page 11, line 7:	Text reads: ‘Ten themes (7 themes concerning the Planning and Implementation Phases; 3 concerning Leadership and Management) emerged from the findings.’ The themes that follow and the information in brackets is confusing. Is it possible to link the themes to the sub-headings more precisely? For instance, 5 for Planning, 3 for Implementation, 6 for leadership and management, as this will correspond directly to the sub-headings that follow. Although there seem to be 14 sub-headings (or themes) and it is not clear how they relate to the 10 themes. There is a need for greater clarity here, which should only require minor changes to the text. In addition, it may be that these sub-headings would benefit from an opening statement to orientate the reader.	We are grateful to the reviewer for highlighting this, and apologise for the confusion. There were 14 themes which emerged from our findings, not 10. We have corrected this in the manuscript and have provided Box 2 as a summary list of the themes.
8	R 1	Page 13, line 21 – 23:	Text reads ‘...indicating some complementary in their disciplinary paradigms such as theories, research methods, and standards.’ Is it possible to be more explicit and offer an example of which theories, methods and standards? As the point of ‘complementarity rather than competition’ is important in this paper, an example would illustrate the authors’ point more robustly.	We are grateful to the reviewer for highlighting this issue, and have added examples on theories, research methods, and standards to illustrate the complementarity in this section (Lines 281 – 285).
9	R 1	Page 19, Line 58:	: Sub-section: ‘Handling disciplinary differences and managing emotions.’ The discussion of emotions in this section is not very strong, although emotional issues are summed up nicely in the Discussion. Can the authors’ supply further detail and insight in the section, i.e. how did these emotions manifest themselves (as serious argument or minor irritation)? What was their level of intensity?	We have provided further details in this sub-section concerning how emotions manifested themselves (Lines 443 – 445).

10	R 1	Page 20, line 48:	Text reads '...cross-disciplinary collaborations were also identified through a group exercise...' Is it possible to indicate what kind of group exercise, such as what it involved etc? There are many aspects in the paper that are likely to be a source of interest, which the authors should not miss.	We have added more information regarding the group exercise in this section (Lines 461 – 465).
11	R 1	page 20, line 55:	Heading 'Developing research networks', page 20, line 55: Is this section about networks or networking relationships? Suggest the authors review for precision, as the quotations do not seem to explicitly relate to networks.	We are grateful to the reviewer for raising this issue and understand the need for clarity. We have revised the heading to “Developing research networks for possible future collaborations” (Line 467) to make clear that this section relates to developing research networks within the consortium. Many of the consortium members were unknown to each other prior to IMPALA but working together during its course enabled them to form strong professional bonds and has allowed for collaborations beyond the consortium.
12	R 1	Page 22, line 54	Is there any evidence you could cite within the text (especially in view of the literature review) that draws out the comparison with mono-disciplinary research?	In our Discussion in the section ‘Allowing time to promote cross-disciplinary activities’, we have added the following: “While less of a consideration in mono-disciplinary research, cross-disciplinary researchers must build mutual understanding and discuss acceptable ways forward.[26-29] Differences across disciplines can be vast, and include: philosophical;[25, 30, 31] measurement standards;[26] framing of concepts;[32] attitudes to theory and practice;[26] the use and understanding of terminology;[24, 25, 30] and expectations of communication and etiquette.[24, 26]” (Lines 544 – 548) Also, in our Discussion, we have a section emphasising the importance of clarity in defining ‘Cross-disciplinary Research’. In ‘Managing expectations and harmonising goals’, we highlighted that “To harmonise goals, frequent discussions and interactions such as information sharing can be helpful, [3, 18] and

				need to be more frequent and intensive than in mono-discipline research.[9]”. (Lines 508 – 510)
13	R 1	Page 23 line 9,	Text opens with: 'As with previous studies....' are there any notable ones that could be mentioned in the text?	We are grateful for this suggestion. There are indeed notable previous studies that reflect our findings. While stating the key findings, we cited these references.
14	R 1	Page 24.	Sub-section: 'Allowing time to promote cross-disciplinary activities', Are the authors able to guide the readers as to how much time (even if only notional) might be required? :	While from this study we cannot quantify how much time might be required to promote cross-disciplinary activities in general, based upon our study we have outlined the key aspects that require time investment to provide the reader with a guide to the expected level of time investment (Lines 537 – 541).
15	R 1	Page 24, end of line 47 –	There or These?	Corrected to “These”, thank you for flagging (Line 560).
16	R 1	Reference 17	– Nursing Outlook?	Corrected to “Nursing Outlook” (Line 686) – many thanks.

Reviewer 2 – Dr Sergi Fàbregues, Universitat Oberta de Catalunya

17	R 2	page 4, lines 49-49	- The study objectives are unclear. The statement on page 4, lines 49-49, that “our study explores the ‘real life’ (...)” is insufficient and not connected to the study methods.	Thank you. We have updated our study objective as: “This qualitative study explores the actions taken to foster CDR in the ‘real life’ situation of a large programme (IMPALA). Our aim is to recommend actions that can be used to improve the effectiveness of future global health CDR programmes.”
----	-----	---------------------	--	---

			a) What are the objectives of the survey? b) What are the goals of the semi-structured interviews? Therefore, a more precise description of the study objectives and a more explicit linkage between these objectives and the methods are needed.	a) This survey’s main objectives were to explore experience and confidence of researchers in IMPALA in doing cross-disciplinary research and to give feedback to the IMPALA management team on possible aspects for further strengthening and improvement in fostering cross-disciplinary research. These main objectives of the survey were however separate from this study. For the study reported here, as we stated in Methods, “The survey findings were used to identify potential interviewees and tailor interview guides.” (Lines 145-146) We also echoed this in the Semi-structured interviews section that “Guided by our baseline survey data 31 primary interviewees were selected using the IMPALA team directory as a sampling frame.” (Lines 151 – 152) b) The goals of the semi-structured interviews were to collect data on challenges and practical actions/solutions related to fostering and conducting CDR in IMPALA (Lines 148 – 149).
18	R 2	Methods	- The description of the study methods lacks detail and clarity. For example, how was the survey data analyzed? a) How were the findings data used to identify the potential participants and build the interview guides? b) What was the rationale for using maximum variation sampling? c) Who carried out the interviews, and when were they conducted? d) What information did the information sheet contain, and why was it provided to the participants? e) In what ways could the information sheet have affected the interview findings? f) How was reflexivity during the interview managed? g) What criteria were followed to ensure trustworthiness?	We thank the reviewer for these questions, which we have addressed individually below: While we used data from the baseline survey in this study for the identification of potential interviewees (based on demographic data), additional analyses of this survey data are beyond the scope of this study; we aim to report these in a separate manuscript to be published later. The following address each of the reviewer’s lettered points: a) For the purpose of this study, we focused the survey findings on “participants’ personal information, and their experience of, and confidence in, conducting CDR” (Lines 144 – 145). We echoed this in the Semi-structured interviews sample selection: “Purposive sampling was used to maximise variation in roles, disciplinary backgrounds, career stages, gender, affiliated organisations,

		 h) How were the data from the different sources triangulated? i) What was the rationale behind triangulation: validating the findings from the various sources; seeking complementarity? j) In what ways was the framework used to analyze the data? 	and geographical locations.” (Lines 152 – 153) Information on the roles, disciplinary backgrounds, career stages, gender, affiliated organisations and geographical locations were derived from the survey findings (Line 154 – 155)  b) We have added the rationale for using maximum variation sampling is “to achieve maximum variation in participants’ characteristics” (Line 154) with a reference to support this rationale. c) We have added “YD, an experienced social scientist with substantial experience of research interviews, carried out the interviews between July 2018 and November 2019.” (Line 157 - 158) d) The information sheet for the interviews/survey was part of fulfilling our ethical obligation to participants. It was to inform potential interview/survey participants why they had been invited to participate, to enable them to make an informed decision regarding their participation in the study. It covered the following aspects: what’s the purpose of the interviews/survey, what is the MUDI project (note: the study reported here was part of the MUDI project), why have you been invited, do you have to take part? What will happen to you if you take part? Compensation, what are the possible benefits of taking part, what will happen to any data you give and to the results of the research study? What will happen if you don’t want to carry on with the study? Will you participant in the interview/survey? If you decide to participant in the interview, may I do an audio-recording of the interview? (note: this question and further explanation applies only to the information sheet for interviews, not for the survey). We have attached the participant information sheet for interviews and for the survey as supplementary file 6 and 2 respectively. e) We consider any potential influence on participant responses from the information sheet to be negligible. As explained in d), the participant information sheet provided high-level background information about the study, informing potential interviewees why they had been invited to participate, in order that they could make an informed decision regarding their participation. We
--	--	--	---

				attach the information sheet as supplementary file 6 for your reference. f) We have added a section 'Reflexivity throughout the interview process' (Lines 168 – 176). “We used reflexivity throughout the interview process to improve the rigour of the data collection. We acknowledged our role as IMPALA members in conducting research on cross-disciplinary working in IMPALA may affect interviewees’ responses which we mitigated by ensuring confidentiality and only sharing unidentifiable summary findings. The interviewer transcribed the first four interviews to familiarise herself with the data and to reflect on the interview process for further improvement. The interviewer also had several debriefing meetings with IB, the senior researcher, on interviews and reflected on how to improve interviews further and on data analysis.” g) To enhance trustworthiness, we used maximum variation sampling to enhance representation of the study population and saturation was achieved. We built trust with our interviewees by ensuring confidentiality; the credibility of our findings is strengthened from having used interview and observational data from diverse interviewees and events, corroborated by document analysis; and the interpretations are clearly placed in and related to the context. More information has been provided in 'Strengths and limitations of the study' (Lines 568 – 582). h) Information from the document review and the observation forms that related to the narrative themes were summarised and compared with these narrative themes to triangulate the findings (Lines 198 – 200). i) The rationale behind triangulation is to enhance trustworthiness of the research findings. When the findings validated one another, we stated it clearly (e.g. Line 443 – 445). When the findings were complementary with each other, we presented them all and clearly described
--	--	--	--	--

				the sources of the findings (Line 368 – 371). j) As we stated in Data Analysis, we used the framework to code, map and analyse the interview data. We also coded data inductively to allow themes emerging from the data. That is why our key themes/findings mainly followed the framework but were slightly different, especially the themes that emerged in the Implementation Phase.
19	R 2	Results	- The authors used a mixed methods design to carry out the study. While the authors don't need to use the mixed methods jargon to describe the methods, they could use the guidelines suggested in that literature to improve the study reporting. Such guidelines include reporting a rationale for using several data sources, describing the type of design used, and making explicit how the integration between the data sources was carried out.	We thank the reviewer for raising this issue. We would like to clarify that this study is a qualitative study using mixed qualitative research methods (rather than using mixed qualitative and quantitative methods). We reported our study following the qualitative reporting guideline, 'Standards for Reporting Qualitative Research'.
20	R 2		- The authors cite a source on narrative analysis (Riessman, 2008) to justify the type of qualitative analysis, although the study was not based on a narrative approach. What did "thematic synthesis" imply in the study?	1) Regarding a narrative analysis approach: This study was based on a narrative approach. We analysed narratives of our interviewees' experiences and views including their explanation and suggestions. We then took a comparative approach to interpreting similarities and differences among participants' narratives. This approach is the same as that described in Riessman, 2008. 2) Regarding thematic synthesis We have provided more information, Lines 195 – 198. Using thematic synthesis, we produced a narrative summary of the extracted information through an inductive and deductive combined approach using a 'constant comparison' method. This involves coding the data and identifying themes and sub-themes, which were then

				adjusted iteratively by constantly comparing among them through reflection and analyses. In this way the themes and sub-themes were refined and integrated to form the basis of a coherent and explanatory descriptive narrative.
--	--	--	--	---

VERSION 2 – REVIEW

REVIEWER	Reeves, Julie University of Southampton, Centre for Higher Education Practice
REVIEW RETURNED	27-Feb-2022

GENERAL COMMENTS	I would like to thank the authors for addressing the points raised in the initial review. I am pleased to see the changes made have noticeably enhanced the paper. By revising the title, amending some of the text and clarifying some of the sections, the paper is, now, a cogent and coherent read. I am looking forward greatly to the publication of this paper. It will serve as a useful resource not only for those working in cross-disciplinary research in health and medical areas, but also as advice to principal investigators, funders and institutions who lead and support international multidisciplinary projects in other areas too.
--